



Formation of highly oxygenated organic molecules from chlorine atom
initiated oxidation of alpha-pinene
Yonghong Wang[1], Matthieu Riva[1,2], Hongbin Xie[3,1], Liine Heikkinen[1], Simon
Schallhart[1], Qiaozhi Zha[1], Chao Yan[1], Xucheng He[1], Otso Peräkylä[1] and Mikael Ehn[1]
[1]Institute for Atmospheric and Earth System Research / Physics, Faculty of Science, P.
O. Box 64, 00014 University of Helsinki, Helsinki, Finland
[2]Univ Lyon, Université Claude Bernard Lyon 1, CNRS, IRCELYON, F-69626,
Villeurbanne, France
[3]Key Laboratory of Industrial Ecology and Environmental Engineering (MOE), School
of Environmental Science and Technology, Dalian University of Technology, Dalian
116024, China

15                  Submitted to: Atmospheric Chemistry and Physics

17                  Corresponding to: Yonghong Wang and Hongbin Xie

18                  yonghong.wang@helsinki.fi; hbxie@dlut.edu.cn



Abstract
Highly oxygenated organic molecules (HOMs) from atmospheric oxidation of alpha-
pinene can irreversibly condense to particles and contribute to secondary organic
aerosol (SOA) formation. Recently, the formation of nitryl chloride ($ClNO_2$) from
heterogeneous reactions, followed by its subsequent photolysis is suggested to be an
important source of chlorine atoms in many parts of the atmosphere. However, the
oxidation of monoterpenes such as alpha-pinene by chlorine atoms has received very
little attention, and the ability of this reaction to form HOM is completely unstudied.
Here, chamber experiments were conducted with alpha-pinene and chlorine under low
and high nitrogen oxide ($NO_x$) conditions. A $NO_3$-based CI-APi-TOF was used to
measure HOM products. Clear distributions of monomers with 9-10 carbon atoms and
dimers with 18-20 carbon atoms were observed under low $NO_x$ conditions. With
increased concentration of $NO_x$ within the chamber, the formation of dimers was
suppressed due to the reactions of peroxy radicals with NO. We estimated the HOM
yields from chlorine-initiated oxidation of alpha-pinene under low-$NO_x$ conditions to
be around 1.8 %, though with a substantial uncertainty range (0.8-4 %) due to lack of
suitable calibration methods. Corresponding yields at high $NO_x$ could not be
determined because of concurrent ozonolysis reactions. Our study demonstrates that
chlorine atoms also initiated oxidation of alpha-pinene and yields low volatility organic
compounds.










## 1. Introduction



Highly oxygenated organic molecules (HOMs) have been identified as key species in
the formation of new atmospheric aerosol particles and secondary organic aerosol (SOA)
(Ehn et al., 2014, 2017; Kulmala et al., 2013; Bianchi et al., 2019). Recently, the
formation of HOMs in the gas phase was described as an autoxidation process of peroxy
radicals ($RO_2$) via multiple intramolecular H atom shifts (Crounse et al., 2013; Jokinen
et al., 2014b; Mentel et al., 2015; Rissanen et al., 2014). Oxygen-containing moieties
such as carbonyl, carboxylic acid and hydroxyl groups can weaken nearby C-H bonds,
making H-abstraction and autoxidation competitive with bimolecular $RO_2$ reactions,
e.g. with NO (Crounse et al., 2013; Praske et al., 2018). Until now, all studies on the
formation of HOMs have focused on reactions initiated by oxygen-containing oxidants
($O_3$, OH and $NO_3$).

Increasing evidence indicates that the chlorine atom (Cl) may also play an important
role in transforming atmospheric organics. (Tham et al., 2016; Thornton et al., 2010).
Chlorine atoms have the greatest reactivity toward volatile organic compounds (VOC),
with rate constants that are, with some exceptions, an order of magnitude higher than
those of hydroxyl radicals (OH) (Riva et al., 2015). Historically, chlorine atoms were
thought to be formed primarily from heterogeneous reaction cycles involving sea salt,
and its concentration estimated to be around 1-10% of that of OH. Therefore, the role
of chlorine atoms in atmospheric oxidation processes has traditionally been thought to
be limited to the marine boundary layer only. In recent years, $ClNO_2$, as a significant
chlorine atom source, was found in continental regions of America, Canada and
Germany, and high concentrations of $ClNO_2$ were also detected in the urban atmosphere
in China (Reyes-Villegas et al., 2018; Tham et al., 2016; Thornton et al., 2010; Wang
et al., 2017). The new findings have expanded the potential importance of chlorine
atoms from coastal areas to continental urban areas. A recent study also reported that
chlorine atoms can be more important than OH radicals for the oxidation of alkanes in



the North China Plain (Liu et al., 2017). Therefore, it is desirable to probe the role of
chlorine radicals in the degradation of VOCs and related SOA formation.

Emission of biogenic volatile organic compounds (BVOC) to the atmosphere
dominates total hydrocarbon emissions on a global scale, with methane, isoprene and
terpenes having the highest source strengths (Guenther et al., 2012). Alpha-pinene is
the most abundant monoterpene in the atmosphere and its oxidation products from
ozonolysis and photooxidation contribute to a substantial fraction of SOA mass
(Riccobono et al., 2014; Zhang et al., 2018). Chlorine atom initiated reactions of alpha-
pinene have also been shown to contribute to the formation of SOA, which implies that
low volatile compounds are efficiently produced also in this process (Cai and Griffin,
2006; Ofner et al., 2013).

Similar to the reaction with OH radicals, the reaction of VOCs with chlorine atoms may
proceed either via addition of Cl to unsaturated bonds or via H-abstraction. Wang et.al
(2017) found that the Cl addition to isoprene can lead to the formation of low volatility
organic compounds. In principle, Cl-initiated reactions could form HOMs in a similar
manner as OH-initiated reactions (Berndt et al., 2016a), as the initial addition or
abstraction step is comparable for both oxidants. In view of the increased understanding
of the importance of chlorine atoms in atmospheric chemistry, it is desirable to
investigate the formation of HOMs from reactions of common atmospheric VOC with
Cl.

Here, a laboratory chamber experiments were performed to investigate the ability of
chlorine atom to form HOMs from the oxidation of alpha-pinene. HOMs were
characterized using a nitrate-based chemical ionization mass spectrometer, under both
low and high $NO_x$ conditions. The yields of these HOMs were determined under the
low $NO_x$ conditions, and the atmospheric implications of this study are discussed.

## 2. Experiment and method
### 2.1 experimental setup
The experiments were conducted in the "COALA" chamber at the University of
Helsinki (Peräkylä et al., 2019; Riva et al., 2019). It is a 2 m$^3$ Teflon chamber, run as a
continuously stirred tank reactor, used with a flow of 45 liter per minutes (LPM),
resulting in an average residence time of about 45 minutes. The chamber is surrounded
by housing to provide dark conditions. No water vapor was added to the chamber, and
the temperature was the same as the temperature of the room, around 25°C. A general
schematic of the chamber facility is shown in Figure 1. Our experiment was aimed to
probe chlorine atom initiated formation of HOMs under low/high NO$_x$ conditions. We
use 400 nm LED lights to photolyze chlorine and NO$_2$ and produce chlorine atoms and
NO as following:

$$NO_2 \rightarrow NO + O$$
$$Cl_2 \rightarrow Cl + Cl$$


The concentration of Cl atoms was varied by changing the amount of 400 nm light. In
practice this was done by turning on different amounts of the available lights, with the
maximum corresponding to seven. We will refer below to the number of lights that were
turned on, although each "light" corresponds to a group of LED strips.

### 2.2 Instrumentation and data analysis

A nitrate ion (NO$_3^-$) based chemical ionization atmospheric pressure interface long
time-of-flight (CI-APi-L-TOF) mass spectrometer was used for measuring HOMs. The
instrument has been shown to be sensitive towards such group of compounds, detecting
them as adducts with the nitrate ion. Due to a lack of suitable calibration methods, the
CI-APi-L-TOF was not calibrated for HOMs during this study. In order to estimate
rough HOM concentrations, we directly use the calibration coefficient (C = 1.6 e10)
utilized by Ehn et al. (2014), which is very close to values utilized in several other



studies using a CI-APi-TOF(Jokinen et al., 2014a; Riva et al., 2019). This approach
obviously brings large uncertainties when the estimating HOM concentrations, which
we estimate to be at least -50 %/+100 %. More detailed information about the
instrument can be found in (Jokinen et al., 2012), noting that compared with the CI-
APi-TOF used before, the long time-of-flight mass spectrometer used here has a
doubled mass resolving power enabling a more accurate assignment of molecular
formulas. Simultaneously, we also used a high-resolution long time-of-flight aerosol
mass spectrometer (HR-L-TOF-AMS) to measure bulk aerosol chemical properties
(Decarlo et al., 2006). As no seed aerosol particles were added to the chamber, the VOC
oxidation products lead to new particle formation and growth to large enough sizes to
be measured by the AMS. We also periodically used a filter in front of the AMS inlet to
see the influence of the background signal to measured aerosol mass concentration.
There were 10 minutes of filter measurements per hour during our experiments. A PTR-
TOF-MS (TOF-8000, Ionicon) was used to measure the concentration of alpha-pinene
in the chamber. The instrument background was determined every day for 20 mins by
guiding the chamber air through a catalytic converter, which removes the VOCs. Then,
the background corrected signals were used to obtain alpha-pinene mixing ratios by
using the calibration coefficient determined before the experiments. A description of
the used setup employed for the calibration and zero air measurements have been
introduced earlier (Schallhart et al., 2018). A custom-built DMPS system was used to
measure the particle number size distribution from 10 nm to 400 nm in the chamber.
The NO concentration was measured with an ECO-PHYSICS CLD 780 TR instrument
with a detection limit of 3 ppt. $NO_x$ (=$NO+NO_2$) concentrations were determined by
using a Thermo-Fisher 42i analyzer. $O_3$ concentration was measured with a Thermo-
Fisher 49i analyzer.

2.2 Estimation of chlorine atom concentrations
During steady state in the chamber, average concentrations of chlorine atom was
calculated using the rate coefficients (4.6±1.3) • $10^{-10}$ $cm^3$ molecule $^{-1}$ $s^{-1}$ of Cl atoms





with alpha-pinene (Finlayson-Pitts et al., 1999), as following:
$$d[AP]/dt = Q\_in - k * [Cl] * [AP] - Q\_out$$
where $Q\_in$ is the concentration of alpha-pinene continuously injected into the chamber,
and $Q\_out$ is the concentration that exited the chamber. The term $Q\_in$ was 13.3 ppb,
while the term $Q\_out$ varied depending on the conditions, and is calculated as $[AP]/45$
min. During steady state, $d[AP]/dt$ is zero, and then $[Cl]$ concentration is calculated
accordingly. As shown in Figure 2, the concentration of HOMs decreased and alpha-
pinene increased as the number of lights switched on changed from 7 to 4, 2 and 1. We
use the variation of alpha-pinene and HOM concentrations during this run to calculate
both chlorine atom concentrations and HOM yields. Each change in alpha-pinene
concentration was due to the change in Cl atom concentration, and with knowledge of
the reaction rate, the concentration of Cl atoms as a function of the number of lights
turned on was determined (Figure 3). The calculated [Cl] concentrations are in the range
of (1-5) x $10^5$ molecules cm$^{-3}$, which is within atmospheric relevant concentration
ranges (Tham et al., 2016). Raw data from the CI-API-L-TOF were recorded in 10s
resolution in HDF format. We used Toftools for data analysis and detailed protocols of
the software have been introduced by Junninen et al. (2010).

## 2.3  HOMs molar yield

The change of HOM concentration with time can be described as follows, in analogy
with Ehn et al. (2014):

$$\frac{d[HOMs]}{dt} = k_1 g[a - pinene][Cl] - k_{loss}[HOMs] \qquad (1)$$


$$g = \frac{k_{loss}[HOMs]}{k_1[a - pinene][Cl]} \qquad (2)$$

Here, $k_1$ is the reaction rate coefficient of alpha-pinene with chlorine atoms and $\gamma$ is the
molar yield of HOMs, i.e., the fraction of alpha-pinene + Cl reactions that produced
HOMs. $k_{loss}$ is the loss rate of HOMs to the chamber walls and particles, though the
latter was negligible in this study due to the low aerosol loadings. We used 300 s as a
lifetime of HOMs, i.e. $k_{loss} = 1/300$ s$^{-1}$, in our previous study in the COALA chamber
(Riva et al., 2019).

## 3.  Results and discussion
### 3.1 Formation of HOMs under low NO condition

Figure 4 (a, b, c and d) shows mass spectra measured by the NO$_3$-CI-APi-TOF during
steady state alpha-pinene oxidation with different amounts of lights switched on. The
x-axis represents mass to charge ratio, in units of Thomson (Th). The y-axis represents
signals in units of counts per second. As we can see, both monomers (280-400 Th) and
dimers (440-580 Th) showed increased signals with increased number of lights, and
consequently increased [Cl]. The most abundant peaks are labeled in Figure 4d, with
some of the largest signals in the monomer range attributed to $C_9H_{12}O_{7,8}$ and
$C_{10}H_{14}O_{8,9,10}$. During the oxidation of $C_{10}H_{16}$, in the absence of NO, the fate of $RO_2$
radicals depends on the concentrations of $HO_2$ and $RO_2$. Autoxidation competes with
bimolecular reactions, becoming more likely at lower $RO_2$ and $HO_2$ concentrations. At
high oxidation rates, autoxidation is likely inhibited compared to actual atmospheric
conditions.

As we show in the Figure 4(d), $C_{10}H_{14}O_8$-$_{12}$ compounds are large peaks in the monomer
range observed with the NO$_3$-CI-API-TOF. These compounds with 14 hydrogens may
come from decomposition of $C_{10}H_{15}O_n$ peroxy radicals via loss of OH or $HO_2$, or
following reactions with other $RO_2$, as depicted schematically in Figure 5. Another



abundant group is $C_{10}H_{16}O_{6-12}$, which may result from $RO_2$ terminated by $HO_2$. In the
dimer range, the most abundant compounds are $C_{19}H_{28}O_{8-14}$ and $C_{20}H_{30}O_{11-14}$. These
compounds come from $RO_2$ cross reactions, as has been shown in multiple earlier
studies. The $C_{20}H_{30}O_n$ dimers are most likely formed from reactions of two $C_{10}H_{15}O_x$
radicals, as were many abundant monomers. As noted earlier, Cl oxidation of alkenes
may occur via a Cl addition (forming an initial radical containing 16 H-atoms and one
Cl atom) or via an H-abstraction reaction (forming a radical with 15 H-atoms and no
Cl) (Figure 5). Based on our HOM spectra, the abstraction reaction would seem to
completely dominate the reaction, as only a few minor peaks in the spectrum could be
identified to contain Cl. Alternatively, only the abstraction pathway leads to HOM
formation, or the Cl atom is lost during the subsequent reaction in the oxidation
processes. With our data, we cannot rule out either of these explanations for this
surprising result. Loss of HCl from alpha-pinene products from Cl oxidation have, to
our knowledge, only been reported to take place in the aerosol phase (Ofner2013).

Figure 6 shows the variation of several closed-shell HOM products and the peroxy
radical $C_{10}H_{15}O_{10}$ measured by $NO_3$-CI-API-TOF when we changed the lights from
dark conditions to 1, 2, 4 and 7 lights switched on. Given the low Cl atom concentration,
it is expected that no multi-generation oxidation by Cl can take place, and the behavior
of all closed shell oxidation products should follow similar patterns. As seen in Figure
6, this was the case both for monomers and dimers. The less steep increase of the radical
is also according to expectations, as the formation of $RO_2$ is linear with the alpha-pinene
oxidation rate, but the loss rate (when dominated by $RO_2$ cross reactions) is proportional
to $(RO_2)^2$. For closed shell species, the wall loss-driven loss rate stays constant
throughout the experiment, and therefore they increase linearly with the alpha-pinene
oxidation rate while the $RO_2$ radicals increase as the square root of the oxidation rate.
For more detailed discussion on $RO_2$ dynamics in a steady state chamber, see Ehn et al.

254    (2014).



In Figure7, we plotted time series of the particle number size distribution and the total
number concentration, together with mass concentrations of particulate chloride and
organics as we changed the number of lights. Particle formation was detected even at
the lowest Cl atom concentration, as indicated by the increases in aerosol number
concentration. An increased in aerosol mass concentration as detected by the AMS only
took place at the two highest Cl atom concentrations, when the particles were able to
grow into a size range measurable by the AMS. Particulate chloride mass
concentrations also increased relatively linearly with the concentration of organics as
we increased the number of lights. The Chl/Org ratio was only around 3 %, suggesting
that the majority of condensed OVOC did not contain Cl atoms. Some part of the
chloride signal may also result from adsorption of HCl to particles.

3.2 Formation of HOMs at high NOx

Anthropogenic emissions have a significant influence on the formation of SOA, to a
large part due to the influence of $NO_x$ on the atmospheric oxidation chemistry (Lee et
al., 2016). In general, the fate of peroxy radicals in chamber experiments can be
dominated by reactions with other $RO_2$, $HO_2$ or NO, depending on the exact conditions.
In our experiments without $NO_x$ addition, $RO_2$ was expected to be the main terminator,
as also supported by the high number of detected ROOR dimers. In the atmosphere, all
of the three mentioned reaction partners may be relevant at the same time. However,
with increased anthropogenic influence, the reaction of $RO_2$ with NO will often become
dominant. Therefore, we added $NO_x$ to the chamber as it allowed for the isolation of
the formation pathways leading to HOMs in cases where NO was the main terminator
for $RO_2$ radicals. Figure 8 depicts a HOM mass spectrum at steady state during alpha-
pinene oxidation by chlorine radicals in the presence of ~10 ppb $NO_x$, with the
maximum 7 lights turned on. As anticipated, the dimers above 440 Th were greatly
reduced compared to the runs without $NO_x$. As more lights were turned on, both the Cl
atoms and NO formation increased, as the 400 nm lights photolyze both $Cl_2$ and $NO_2$.





This coupling, together with the fact that the NO₂ photolysis leads to ozone formation,
which subsequently can react with alpha-pinene to form HOMs, limits our quantitative
analysis of these experiments. However, we conclude that efficient HOM formation
took place also under these high-NOx conditions, and thus the autoxidation occurs
rapidly enough to still compete with $RO_2$ termination reactions. The $NO_x$ addition also
formed an abundance of organonitrate compounds like $C_{10}H_{15}NO_{8,9,10,11,12}$, as shown in
Figure 8. This family of compounds may form from H-abstraction by the chlorine
radical, followed by autoxidation and finally radical termination by NO. The concurrent
formation of ozone means that also some alpha-pinene ozonolysis reaction will take
place, though oxidation by Cl atoms was still the main loss for alpha-pinene also under
these conditions.

Figure 9 shows variation of some nitrogen-containing HOMs and variation of alpha-
pinene, ozone, NO and $NO_x$, as we changed the lights from dark conditions to 1, 2, 4
or 7 lights switched on. The concentrations of alpha-pinene and NO₂ decreased because
of the consumption by chlorine radicals and photolysis of NO₂ into NO. Importantly,
we did not observe any SOA when we had NO in the chamber. NO may have suppressed
the particle formation by suppressing the dimer formation, as these have been shown to
be important for initial particle formation (Tröstl et al., 2016).

3.2 Estimated HOMs production yields

Quantifying the molar yields of HOMs is essential to know their potential importance
from a specific system. We attempt to estimate the molar yield in the case of Cl
oxidation of alpha-pinene in the absence of $NO_x$. The initial $C_{10}H_{16}$ concentration is
around 13.3 ppb without any UV lights switched on in the chamber. As we changed the
lights, alpha-pinene and HOM concentrations varied as we showed in Figure 3. In
addition, we calculated the concentration of Cl radicals as introduced in the Methods
section. With this information, we can calculate the formation rate of HOM, which in





steady state equals the HOM loss rate [HOM]*$k_{loss}$. We can also calculate the oxidation
rate of alpha-pinene as [alpha-pinene]*[Cl]*$k_{AP+Cl}$. The ratio of these two numbers
corresponds to the HOM molar yield. We selected the same runs as in Fig. 3, used also
for calculating the chlorine radical concentration, and calculated the ratio as a linear fit
to these four conditions (Figure 10). We get a slope of 0.018, meaning a HOM yield of
1.8%. Considering the uncertainty in estimating absolute HOM concentrations, we
conservatively estimate that the molar HOM yield from alpha-pinene + Cl is within the
range of 0.8-4 %. These values are similar to HOM yields reported for alpha-pinene
oxidation by ozone and OH (Berndt et al., 2016; Ehn et al., 2014).

4.Conclusion

We have systematically explored the reactions of alpha-pinene with chlorine atoms in
a simulation smog chamber under atmospherically relevant conditions. We measured
substantial amounts of highly oxidized organic molecules (HOM) with a $NO_3$-CI-API-
TOF. With increasing UV lights, and consequently higher chlorine radical
concentrations, the concentrations of both HOM and secondary organic aerosol
increased. With addition of NOx, HOM monomer formation was still efficient, but the
particle formation decreased greatly. We estimated HOM molar yields of around 1.8 %
(0.8-4 %) from the reaction of alpha-pinene with Cl atoms. Our study thus indicates
that in regions where chlorine atom oxidation is of importance, its possible reactions
with monoterpenes can be an important source of HOM, and consequently, SOA.

**Acknowledgement**

338       This work is supported by European Research Council (Grant 638703-COALA)

project and Academy of Finland, via the Center of Excellence in Atmospheric Sciences
and project numbers 317380 and 320094. We acknowledge the Toftools team for



providing the software.

**Competing financial interests**
The authors declare no competing financial interests.

**Author contributions**
Y. H. W, H. B. X and M. E had the original idea of the study. Y. H. W, M. R and H. B.
X conducted the chamber experiments. Y. H. W, M. R, H. B. X, L.H and M. E
interpreted the data. Y.H.W plotted the figures, wrote the manuscript with comments
and suggestions from all co-authors.














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



563                              Figure and table captions



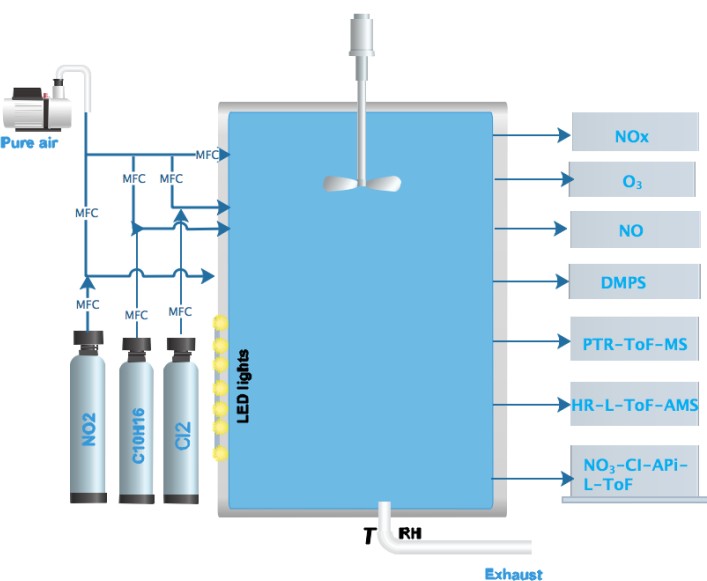

Figure 1 A schematic of the chamber setup and instruments used in the experiment.















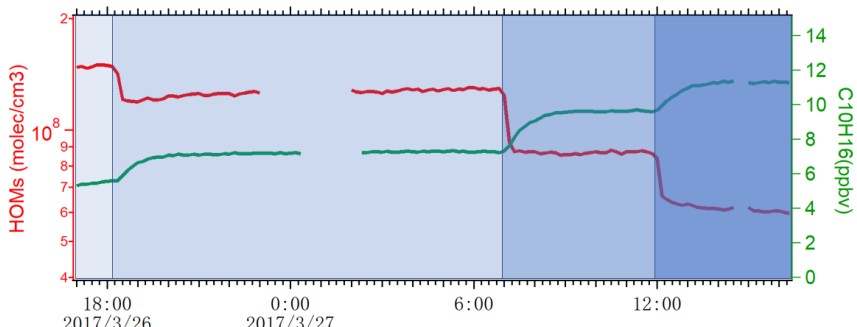


Figure 2. The variation of total HOM concentration and alpha-pinene during four
experiments where the 400 nm lights were decreased stepwise from 7 lights to 4, 2

582                             and 1 light, respectively.













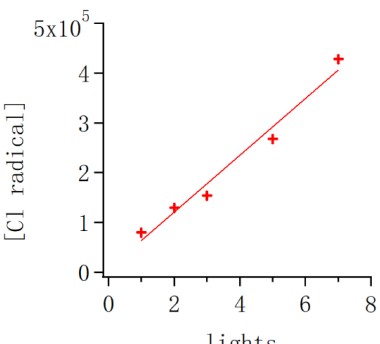


Figure 3. The variation of chlorine radical concentration as a function of lights. The

593          input alpha-pinene concentration was kept constant throughout the experiments.


















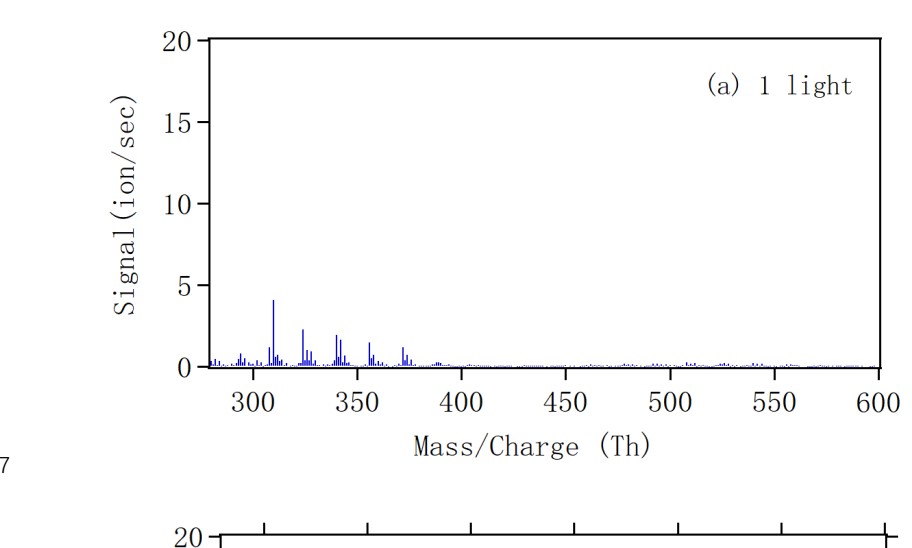


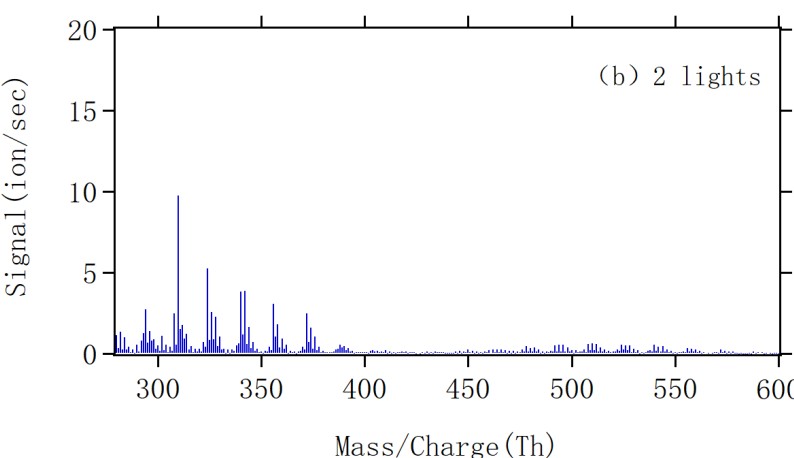


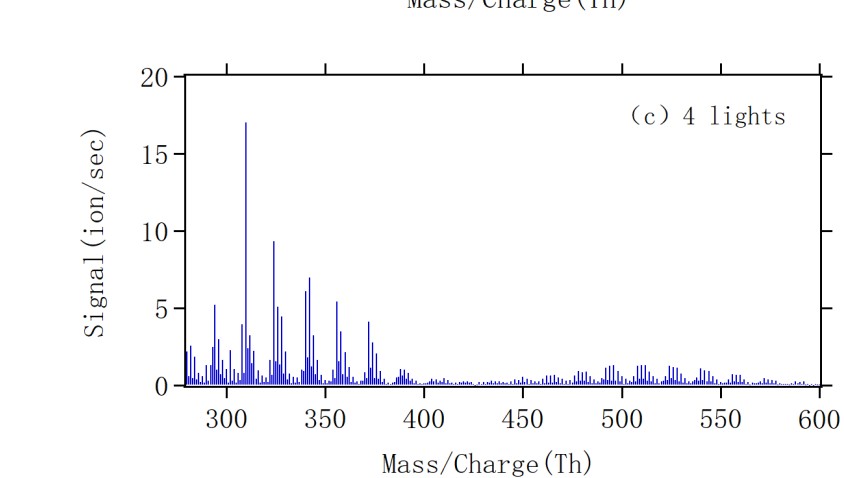






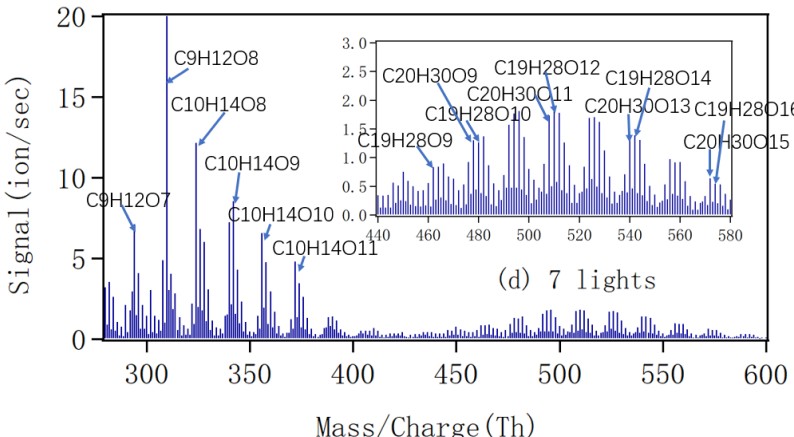

Figure 4. The mass spectra obtained by NO3-CI-API-TOF during steady state with 1(a), 2(b), 4(c) and 7(d) lights. All peaks are detected as clusters with NO3-. The spectra are plotted as unit mass resolution, with background signals removed, but the peak identifications (labeled in panel d) are based on high resolution analyses. The spectra correspond to the same four steady state conditions depicted in Fig. 2.





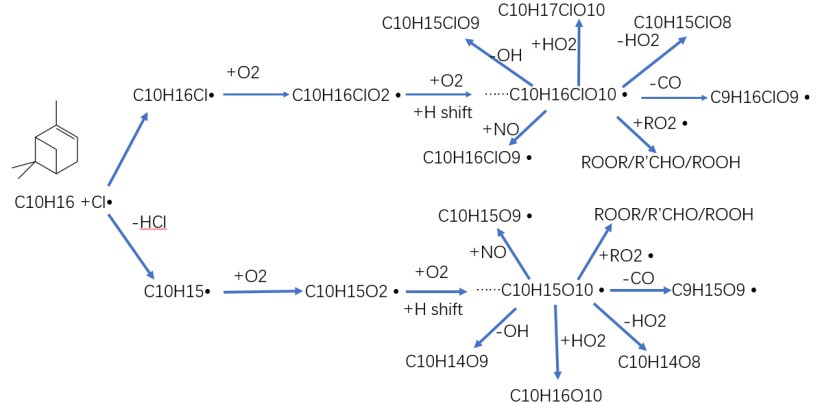

Figure 5. Proposed chemical pathways for chlorine radical oxidation of alpha-pinene, and subsequent autoxidation and HOM formation. The upper path shows the chlorine radical addition pathway, while the lower chain shows hydrogen atom abstraction pathway. In both cases, initially a C-centered radical forms ($C_{10}H_{16}Cl$ or $C_{10}H_{15}$) to which $O_2$ adds to form an initial peroxy radical. This peroxy radical may then undergo multi-step autoxidation to reach the example molecules $C_{10}H_{16}ClO_{10}$ or $C_{10}H_{15}O_{10}$ before termination. The observed HOM spectra in this study suggest a completely dominant role of the lower, H-abstraction, pathway for HOM formation.


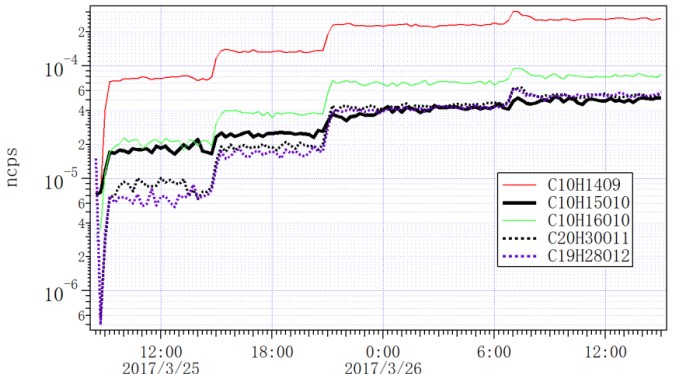



Figure 6. Time series of selected closed-shell HOM monomers, dimers and an RO$_2$
radical (C$_{10}$H$_{15}$O$_{10}$) detected by NO$_3$-CI-APi-TOF as the lights increased from 0 to 1,
2, 4 and 7.














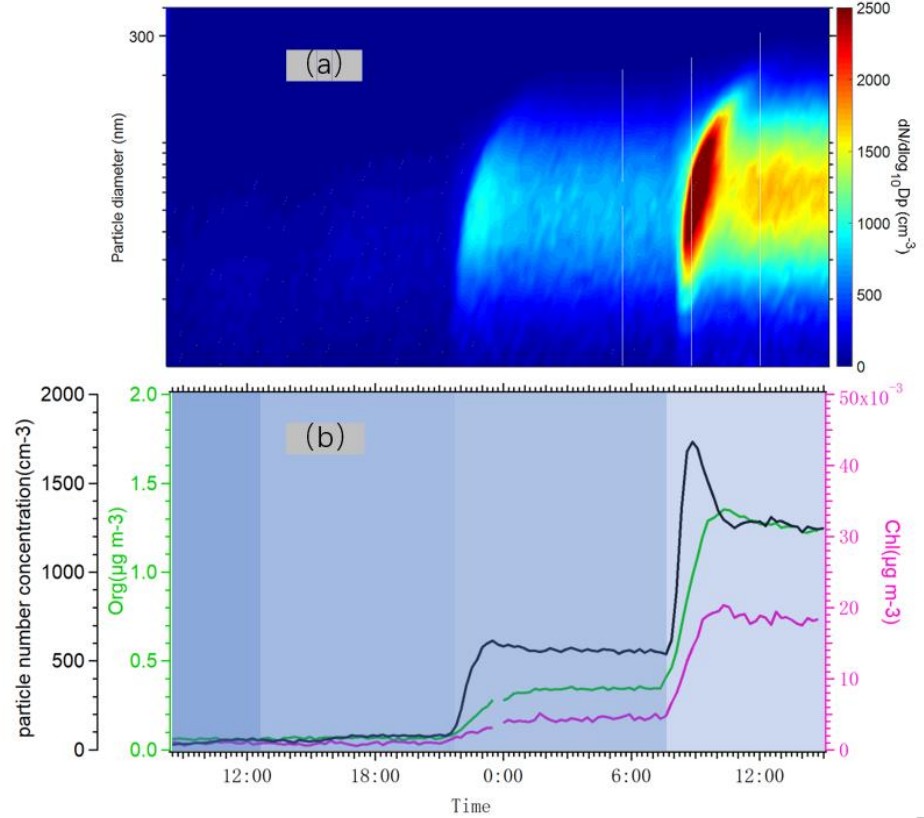



Figure 7. (a) Particle number size distribution measured by DMPS from 10 nm to
400nm when the lights varied from 1 to 2, 4 and 7. (b) Time series of total number
concentration (black) measured by DMPS, organic aerosol concentration (green) and
particulate chloride concentration (pink)measured by AMS.






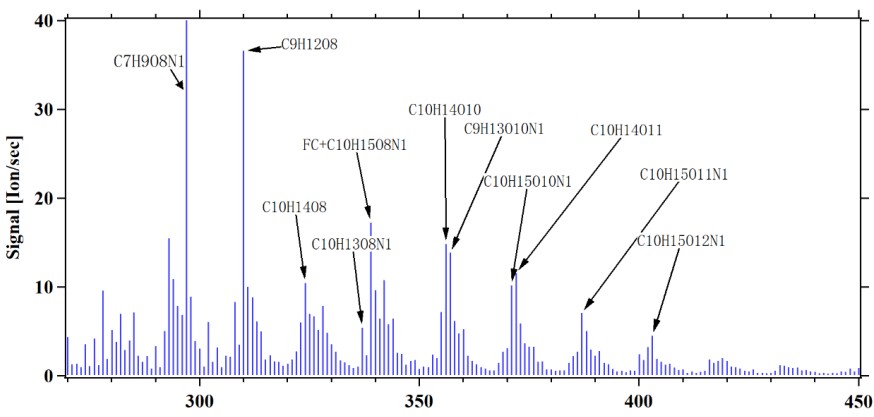


Figure 8. HOM mass spectrum during steady state alpha-pinene oxidation in the presence of 10 ppb NOx, with 7 lights switched on. In addition to molecules detected also in the experiments without NOx, several abundant organ nitrate peaks are formed. Note that a fluorinated compounds (FC) overlaps with the organ nitrate $C_{10}H_{15}O_8N$ at 339Th. All peaks are detected as clusters with $NO_3^-$.













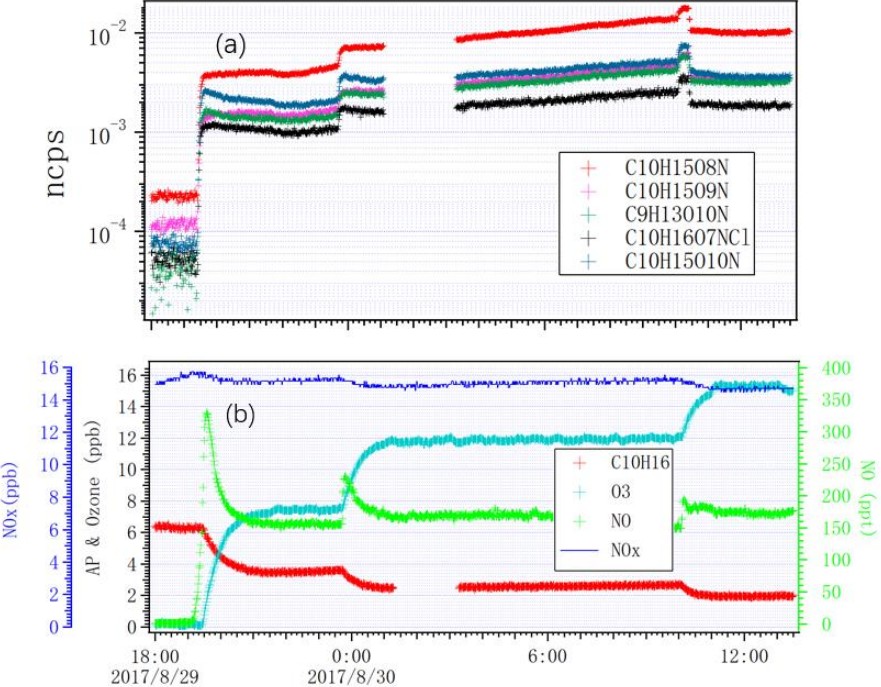



Figure 9. Time series of (a) selected HOMs measured by NO₃-CI-L-API-TOF and (b)
NOₓ, a-pinene, ozone and NO, as the lights switched on from zero to 2, 4, and 7.












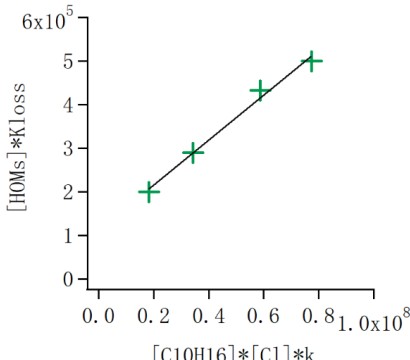



Figure 10. HOM loss rate ([HOM]*Kloss) as a function of the alpha-pinene
oxidation rate. In steady state the loss rate equals the formation rate, and thus
the slope of the points gives the molar yield of HOM from the alpha-pinene +
Cl reaction. The data corresponds to the conditions with 1, 2, 4 and 7 lights
switched on in the chamber, respectively. The slope indicates an average molar
yield of HOMs of 1.8%.

