# Peer review of "Formation of highly oxygenated organic molecules from chlorine atom initiated oxidation of alpha-pinene"

_Atmospheric Chemistry and Physics, 2019_

## Referee Comment (RC1) · Anonymous Referee #2 · 5 Nov 2019

General comments: This work by Wang et al. describes gas-phase measurements of highly oxygenated organic molecules (HOM) from alpha-pinene + Cl oxidation. The study was motivated by recent findings that ClNO2 could be an important source of Cl in certain regions. Chamber experiments were performed to study alpha-pinene + Cl reactions with and without NOx. HOM were measured using NO3–CI-APi-TOF. The experiments, measurements and analysis are all using the well-established HOM study approach while looking at a new reaction system. The result of a HOM yield of 1.8 (0.8 – 4)% adds this new system to the HOM-forming family. Overall, the manuscript is well written and demonstrates new findings regarding HOM formation chemistry, and thus should be eventually published at ACP. But a few major concerns need to be addressed

first.

Specific comments: 1. Line 139-142. Elaborate on this statement. What does this "C" value mean? How do you estimate HOM concentration (ppt) from measured in $NO_3$-CIMS (cps) using this coefficient? This has been described in prior work as referred, but as a standalone article, it should be clarified, at least briefly. Also, how big of uncertainty might be caused by using this value directly? Presumably, the rough calibration was based on $H_2SO_4$. I'm curious whether different instrument configurations (i.e., normal TOF vs. LTOF) could lead to different calibration/quantification results. I'm sure that authors or other users have used the LTOF to calibrate HOMs using $H_2SO_4$. I suggest that the authors at least do such rough calibration and constrain the calibration coefficient. A substantial portion of the conclusion is based on this assumption and simply saying the instrument was not calibrated due to the lack of calibration methods is not satisfactory.

2. Through personal communication with other LTOF users, it was suggested that the APi-LTOF could leak to more molecular fragmentation in comparison to APi-TOF. Is this consistent with what the authors observed in this study? If so, would that lead to ambiguity interpreting monomer vs. dimer HOMs?

3. There are a few errors in Line 173-177. In the mass balance equation (which also needs an equation index), the unit of each term should be concentration/time (e.g., ppb/min). Thus, $Q_{in}$ and $Q_{out}$ do not represent concentrations, but the flow rate/volume. Make corrections accordingly.

4. In Figure 4, it seems the largest monomer is a C9 product ($C_9H_{12}O_8$). Do the authors have an idea of the structures or formation mechanisms of the $C_9H_{12}O_x$ products? Also, the monomer HOM measured here are unlike the other alpha-pinene oxidation studies (e.g., alpha-pinene + $O_3$, HOM dominated by $C_{10}H_{16}O_x$; alpha-pinene + OH, HOM dominated by $C_{10}H_{18}O_x$). Here the HOM are dominated by $C_{10}H_{14}O_x$. This is very interesting. I suggest the authors discuss the mechanism in this route.

5. Line 247-250. To make this argument sufficiently, it probably needs the timeseries of more than just one RO2. Also, quadratic relationship should be more clearly shown by plotting RO2s against reacted alpha-pinene (like Figure 1 in Zhao et al., 2018 PNAS).

6. Line 264-265. This interpretation is not accurate. The AMS measures only the elemental composition. Let's do the math here for example, if a particle-phase molecule of a Mw of 300 has one Cl atom (Mw 35.5), then the Cl/Org $\sim$ 13%. A measured Cl/Org $\sim$ 3% suggests about 23% of the molecules contain Cl on average. This is not a trivial fraction. Of course, this number could vary if the average Mw of particle-phase molecules change, which the authors could provide better estimates. The point is that 3% Cl/Org does not mean Cl contribute little to SOA. If considering that Cl addition could also lead to products without Cl through fragmentation, it seems the abstraction: addition occurs at 3:1 or lower fractions. Abstraction is still larger, but one cannot rule out Cl addition. It might not produce HOM, but definitely contribute to SOA formation.

7. HOM formation under high NOx. More information should be provided in this section. For example, how much O3 are formed? Given the reaction rates with ïA̧ą-pinene, the authors could also constrain the fractions of alpha-pinene reacting with O3 vs. Cl. If alpha-pinene + O3 is the dominant process, then the HOM could be mostly from alpha-pinene + O3 and the statement in Line 287-289 might be wrong. Alternatively, if alpha-pinene + O3 is not the dominant process for alpha-pinene, then the author should consider adding alpha-pinene + O3 in equation (1) and (2) and split HOM contribution from both oxidants. In this way, if the HOM yield from alpha-pinene + O3 is known based on the authors' prior work, then the HOM yield from alpha-pinene + Cl in the presence of NOx could be estimated as well. I'm also curious whether the monomeric range mass spectrum as shown in Figure 4 changes when NOx in present (e.g., different dominating patterns).

8. The most interesting finding to me in this manuscript is that Cl oxidation of alpha-pinene occurs mostly via abstraction, rather than Cl addition. As a general comment, I feel the authors should add some discussion of this preference. Has this been reported

in prior studies? If not, then the authors should point out that this mechanistic behavior needs to be studied in detail in future work.

Technical comments: 1. Line 39 in Abstract. Change "NO3-based" to "NO3–based" and define "CI-APi-TOF".

2. Line 165. NOx = NO + NO2 should be defined earlier in the manuscript.

3. Line 229-230. References are needed.
* * *

---

## Referee Comment (RC2) · Anonymous Referee #1 · 12 Dec 2019

General comments:

Wang et al. report results of laboratory chamber experiments of chlorine-atom-initiated oxidation of a-pinene in low and high NOx conditions. They conduct the experiments in a steady state chamber, varying the number of lights that are turned on to simulate different oxidation rates. They use published a-pinene+Cl reaction rates to calculate the concentration of Cl radicals in the chamber at different number of lights on, and report that the Cl concentrations are similar to atmospheric conditions. For low NOx conditions, new particle formation (based on number concentrations) is reported for all light intensities, but an increase in mass concentrations was only detected by the

[Figure]

AMS when 4 and 7 lights were on. Under high NOx conditions, dimer formation was suppressed and no SOA was detected; ozone was also formed which may have resulted in concurrent a-pinene ozonolysis reactions due to which the authors decided not to calculate HOM yields under high NOx conditions. HOM production yields from the low NOx conditions are calculated as 1.8% (within 0.8-4%), with high uncertainty due to a lack of calibration standards for HOM. Overall, the manuscript is well written, and describes results from experiments designed to address a scientific question not previously answered. I recommend publication of the manuscript after the comments below have been addressed.

Specific comments:

1) Experiments were conducted without the addition of water vapor, resulting (I assume) in an RH of $\sim$ 0% (please clarify in the manuscript if this is the case). As such RH is not a realistic condition, I request that the authors discuss how results (i.e. the formation of HOM and SOA) are expected to change under RH conditions that are more atmospherically relevant.

Section 1, Introduction:

2) Please provide references for: the studies that have focused on HOM formation from O3, OH and NO3; and for the concentration of Cl atoms being estimated as 1-10% of OH radicals.

3) It may be beneficial to discuss in which urban settings the reaction of a-pinene and Cl would be relevant, and express what range of RH would be encountered there.

Section 2.2, Instrumentation and analysis:

4) Please provide more information on how Ehn et al. (2014) obtained the HOM calibration factor, and whether or not it is expected to be instrument-specific. Why is the stated uncertainty range -50% to +100%? Could it be much larger, e.g. could the sensitivity differ by a factor of 10? If not, why not?

5) lines 220-221: Clarify why at high oxidation rates, autoxidation is thought to be inhibited compared to atmospheric conditions. Does this work support that conclusion, and if so, how?

6) lines 229-230 – please provide reference for 'earlier studies'

7) lines 264-266 - Wang and Hildebrandt Ruiz (2017) suggested that organochlorides may not flash vaporize efficiently, leading to an underestimation of the chloride content in the aerosol. While this earlier work was based on an aerosol chemical speciation monitor, the aerosol mass spectrometer used here uses the same vaporizer. I suggest the authors discuss this possibility and analyze the relative abundance of the Cl+ vs. the HCl+ signal in the particle spectra as suggested by Wang and Hildebrandt Ruiz.

Reference:

D. S. Wang and L. Hildebrandt Ruiz*. Secondary organic aerosol from chlorine-initiated oxidation of isoprene, Atmospheric Chemistry and Physics, 13491-13508, 17, 2017.

Section 3.2:

8) Are chlorinated organonitrates observed?

9) lines 293-295: How do the authors reach the conclusion that Cl oxidation (vs. ozonolysis) is the main loss mechanism for a-pinene? It should be possible to calculate this based on estimated Cl concentrations, measured O3 concentrations and published reaction rates, but this calculation is not mentioned in the manuscript.

10) lines 632-633: earlier in the manuscript the authors offer alternative explanations for the lack of Cl species in the HOM spectra. It is my understanding that the data presented here are not sufficient to conclude that H-abstraction is the dominant pathway. I suggest removing or modifying this sentence. Was HCl measured, and could its abundance point to the relative importance of H-abstraction vs. Cl- addition pathways?

Technical comments:

line 73: remove period after "organics"

line 78: "their" instead of "its"

line 101: et al. (instead of et.al)

line 110: remove "a" before laboratory

line 117: capitalize subtitle?

line 138: "this" instead of "such"

line 169: there are two sections 2.2

line 240: citation formatting

line 260: replace 'increased' with 'increase'

line 305: there are two sections 3.2

line 677-678: organic nitrates*
* * *

---

## Author Comment (AC1) · 13 Feb 2020

**A point to point response to the reviewers' comments**

We thank the three reviewers for their comments, and we think their comments and suggestions improved our manuscript. Here are points to points responses (in blue colored), accordingly, we also revised manuscript (in blue colored).

General comments: Wang et al. report results of laboratory chamber experiments of chlorine-atom-initiated oxidation of a-pinene in low and high NOx conditions. They conduct the experiments in a steady state chamber, varying the number of lights that are turned on to simulate different oxidation rates. They use published a-pinene+Cl reaction rates to calculate the concentration of Cl radicals in the chamber at different number of lights on, and report that the Cl concentrations are similar to atmospheric conditions. For low NOx conditions, new particle formation (based on number concentrations) is reported for all light intensities, but an increase in mass concentrations was only detected by the AMS when 4 and 7 lights were on. Under high NOx conditions, dimer formation was suppressed and no SOA was detected; ozone was also formed which may have resulted in concurrent a-pinene ozonolysis reactions due to which the authors decided not to calculate HOM yields under high NOx conditions. HOM production yields from the low NOx conditions are calculated as 1.8% (within 0.8-4%), with high uncertainty due to a lack of calibration standards for HOM. Overall, the manuscript is well written, and describes results from experiments designed to address a scientific question not previously answered. I recommend publication of the manuscript after the comments below have been addressed.

Response: we thank the reviewer for the positive comments. We think these comments are really important to improving the manuscript.

Specific comments: 1) Experiments were conducted without the addition of water vapor, resulting (I assume) in an RH of ~ 0% (please clarify in the manuscript if this is the case). As such RH is not a realistic condition, I request that the authors discuss how results (i.e. the formation of HOM and SOA) are expected to change under RH conditions that are more atmospherically relevant.

Response: Yes, the experiment was conducted under RH around 0%. We do not expect to see an influence of RH on HOM formation, as also indicated by an earlier study (Li et al., (2019). SOA mass did not change under RH from 3%~92%, although particle number concentration has decreased 2~3 time as RH decreased. We added a short discussion on this ( line 127-130).

Section 1, Introduction: 2) Please provide references for: the studies that have focused on HOM formation from O3, OH and NO3; and for the concentration of Cl atoms being estimated as 1-10% of OH radicals.

Response: We added references in the revised version (Line 70;78).

3) It may be beneficial to discuss in which urban settings the reaction of a-pinene and Cl would be relevant, and express what range of RH would be encountered there.

Response: In the paragraph on lines 71-93, we describe several continental regions where Cl has been identified to be an important oxidant. It is likely that many of these will also have BVOC emissions, but we prefer to not narrow down geographically the locations where these reactions will be the most important. We added text to lines 96-98 discussing this issue.

Section 2.2, Instrumentation and analysis:

4) Please provide more information on how Ehn et al. (2014) obtained the HOM calibration factor, and whether or not it is expected to be instrument-specific. Why is the stated uncertainty range -50% to +100%? Could it be much larger, e.g. could the sensitivity differ by a factor of 10? If not, why

not?

Response: Although the calibration factor is instrument dependent, reported values typically vary from ~$10^9$ to ~$10^{10}$ molec cm$^{-3}$. We have added additional references to the text on this topic, and added the text "Ehn et al obtained their calibration factor from a permeation source using a perfluorinated acid, and also showed through calculations that there were theoretical limitations for having a much larger or a much smaller value for C."

The in-depth discussion on this is found in the methods section of Ehn et al 2014.

5) lines 220-221: Clarify why at high oxidation rates, autoxidation is thought to be inhibited compared to atmospheric conditions. Does this work support that conclusion, and if so, how?

Response: The reaction channel between RO2 and RO2 will be enhanced with the presence of increased oxidants (e.g., OH, Cl atom), and as stated on lines 232-234, "Autoxidation competes with bimolecular reactions, becoming more likely at lower RO2 and HO2 concentrations.". However, we have removed the statement mentioned by the reviewer in the revised version, since we did not have such high oxidant concentration in the experiment.

6) lines 229-230 – please provide reference for 'earlier studies'

Response: We have added reference there in the revised version (Line 247).

7) lines 264-266 - Wang and Hildebrandt Ruiz (2017) suggested that organochlorides may not flash vaporize efficiently, leading to an underestimation of the chloride content in the aerosol. While this earlier work was based on an aerosol chemical speciation monitor, the aerosol mass spectrometer used here uses the same vaporizer. I suggest the authors discuss this possibility and analyze the relative abundance of the Cl+ vs. the HCl+ signal in the particle spectra as suggested by Wang and

Hildebrandt Ruiz. Reference: D. S. Wang and L. Hildebrandt Ruiz*. Secondary organic aerosol from chlorine-initiated oxidation of isoprene, Atmospheric Chemistry and Physics, 13491-13508, 17, 2017. Section 3.2:

Response: The slow evaporating compounds are most certainly a larger issue with the ACSM compared to a TOF-AMS, mainly due to dissimilarities between their mass analyzers (rather *slow* RGA quadrupole vs a *fast* TOF). However, it is important to keep in mind that such effect would likely not be fully negligible in the TOF-AMS either. The figure below visualizes how the organic-to-chloride ratio decreases slightly over time during an experiment conducted between March 21st and 22nd in 2017 (Figure 1a). The decrease is not as significant as observed by Wang and Hildebrandt-Ruiz et al., (2017; Figure 4), and indeed, the $R^2$ is nearly negligible. Notably there was a major difference in mass concentrations, as well as aerosol precursors, between the Wang and Hildebrandt-Ruiz (2017) experiment and ours. We also visualized the temporal behavior of the m/z 35 –to- m/z 36 ratios (Figure 1c; equivalent to $Cl^+$ vs $HCl^+$ -ratio) over the same time period, revealing a nearly negligible increase. We chose not to proceed with a more detailed investigation of the temporal behaviors of chloride-related fragments to narrow down the effect of the slow evaporating organochlorides from the AMS vaporizer, mainly due to the fact that we are not aiming on quantifying the SOA yield nor Chloride yield from these experiments. A main reason for not including those yields arose from the small mass loadings detected (note the chloride mass concentration is only tens of nanograms per cubic meter that is close to limit of detection), and we cannot assure that all the mass was detected by the AMS (no seed aerosol was used make sure the aerosol size distribution was at the AMS measurement range!). Importantly, we made it clear in the manuscript that we are not claiming proper organochloride quantification in the particle phase. Line

[Figure]

**Figure R1** The time series of the organic-to-chloride ratio (panel a), organic and chloride (panel b), m/z 35 –to- m/z 36 ratio (panel c), and *m/z* 35 and *m/z* 36 (panel d) during the experiment on March 21$^{st}$ to 22$^{nd}$, 2017. The time period selected for the figures was when the organic aerosol mass concentration was closest to steady state.

8) Are chlorinated organ nitrates observed? 9) lines 293-295: How do the authors reach the conclusion that Cl oxidation (vs. ozonolysis) is the main loss mechanism for a-pinene? It should be possible to calculate this based on estimated Cl concentrations, measured O3 concentrations and published reaction rates, but this calculation is not mentioned in the manuscript.

Response: Thanks for the comment. Yes, we observed chlorinated organonitrates as shown in figure 9(a), C10H16ClO7N was observed. We reached the conclusion after the calculation as following: [ap_ozone]/[ap_Cl]=[AP]*[Ozone]*2.5e10*$k_{ap+ozone}$/[AP]*[Cl]*$k_{ap+Cl}$, the results is around 0.2. with the results, we conclude that AP reaction with chlorine atom is the main reaction during the system. We added explanations in the revised version (Line: 312-315).

10) lines 632-633: earlier in the manuscript the authors offer alternative explanations for the lack of Cl species in the HOM spectra. It is my understanding that the data presented here are not sufficient to conclude that H-abstraction is the dominant pathway. I suggest removing or modifying this sentence. Was HCl measured, and could its abundance point to the relative importance of H-abstraction vs. Cl- addition pathways?

Response: We removed the statement as suggested. The HCl can unfortunately not be measured by NO3-CI-APi-TOF, and we expect it would require a very accurate measurements to be able to use it as a proxy for the importance of H-abstraction vs. Cl -addition pathway.

Technical comments:

line 73: remove period after "organics"

Response: corrected.

line 78: "their" instead of "its"

Response: corrected.

line 101: et al. (instead of et.al)

Response: corrected.

line 110: remove "a" before laboratory

Response: corrected.

line 117: capitalize subtitle?

Response: corrected.

line 138: "this" instead of "such"

Response: corrected.

line 169: there are two sections 2.2

Response: corrected.

line 240: citation formatting

Response: corrected.

line 260: replace 'increased' with 'increase'

Response: corrected.

line 305: there are two sections 3.2

Response: corrected.

line 677-678: organic nitrates*

Response: corrected.

**General comments:** This work by Wang et al. describes gas-phase measurements of highly oxygenated organic molecules (HOM) from alpha-pinene + Cl oxidation. The study was motivated by recent findings that ClNO2 could be an important source of Cl in certain regions. Chamber experiments were performed to study alpha-pinene + Cl reactions with and without NOx. HOM were measured using NO3–CI-APi-TOF. The experiments, measurements and analysis are all using the well-established HOM study approach while looking at a new reaction system. The result of a HOM yield of 1.8 (0.8 – 4)% adds this new system to the HOM-forming family. Overall, the

manuscript is well written and demonstrates new findings regarding HOM formation chemistry, and thus should be eventually published at ACP. But a few major concerns need to be addressed first.

Response: we thank the reviewer for the positive comments. We think these comments are helpful to improving the manuscript. Accordingly, we revised the manuscript according to your comments.

Specific comments:

1. Line 139-142. Elaborate on this statement. What does this "C" value mean? How do you estimate HOM concentration (ppt) from measured in NO3-CIMS (cps) using this coefficient? This has been described in prior work as referred, but as a standalone article, it should be clarified, at least briefly. Also, how big of uncertainty might be caused by using this value directly? Presumably, the rough calibration was based on H2SO4. I'm curious whether different instrument configurations (i.e., normal TOF vs. LTOF) could lead to different calibration/quantification results. I'm sure that authors or other users have used the LTOF to calibrate HOMs using H2SO4. I suggest that the authors at least do such rough calibration and constrain the calibration coefficient. A substantial portion of the conclusion is based on this assumption and simply saying the instrument was not calibrated due to the lack of calibration methods is not satisfactory.

Response: We have added additional references to the text on this topic, and added the following text to the manuscript to better clarify the calibration factor: "Due to a lack of suitable calibration methods, the CI-APi-TOF was not calibrated for HOMs during this study. In order to estimate rough HOM concentrations ([HOM]), we directly use the calibration coefficient C = 1.6 e10 molec cm$^{-3}$ utilized by Ehn et al. (2014), to

convert the measured HOM ion signals according to the equation below (Jokinen et al., 2014).

$$[HOM] = C * \frac{HOM \cdot NO_3^-}{NO_3^- + HNO_3 \cdot NO_3^-}$$

This value of C is very close to values utilized in several other studies using a CI-APi-TOF (Jokinen et al., 2014, 2015; Riva et al., 2019). Ehn et al (2014) obtained their calibration factor from a permeation source using a perfluorinated acid, and also showed through calculations that there were theoretical limitations for having a much larger or a much smaller value for C." More in-depth discussion on this is found in the methods section of Ehn et al 2014. Importantly, the quantification utilizes a normalization to the reagent ion count rate, which means that most instrumental parameters "cancel out". We are not aware of any reason why or how a different mass analyzer (e.g. HTOF vs LTOF) would impact the value of C. The exact tuning of each instrument can impact the mass-dependent transmission, but this issue should be equally valid independent of mass analyzer.

2. Through personal communication with other LTOF users, it was suggested that the APi-LTOF could leak to more molecular fragmentation in comparison to APi-TOF. Is this consistent with what the authors observed in this study? If so, would that lead to ambiguity interpreting monomer vs. dimer HOMs?

Response: Like the transmission, the fragmentation in the CI-API-TOF is highly related with the turning of the instruments. We have no experience that LTOF would lead to more molecular fragmentations than HTOF. If anything, the opposite might sometimes be true based on LTOF instruments often seeing many more water clusters under higher RH conditions.

3. There are a few errors in Line 173-177. In the mass balance equation (which also needs an equation index), the unit of each term should be concentration/time (e.g., ppb/min). Thus, $Q\_in$ and $Q\_out$ do not represent concentrations, but the flow rate/volume. Make corrections accordingly.

Response: Thanks for the carefully reading, we have corrected the statements in our revised version.

4. In Figure 4, it seems the largest monomer is a C9 product (C9H12O8). Do the authors have an idea of the structures or formation mechanisms of the C9H12Ox products? Also, the monomer HOM measured here are unlike the other alpha-pinene oxidation studies (e.g., alpha-pinene + O3, HOM dominated by C10H16Ox; alpha-pinene + OH, HOM dominated by C10H18Ox). Here the HOM are dominated by C10H14Ox. This is very interesting. I suggest the authors discuss the mechanism in this route.

Response: We can of course only speculate on exact mechanisms, given the data available to us. But we added such discussion on lines 232-235: In addition, the major HOM from a-pinene + O3 are typically also C10H14Ox, though C10H16Ox HOM are also abundant, especially in cases where an OH scavenger was not employed.

5. Line 247-250. To make this argument sufficiently, it probably needs the timeseries of more than just one RO2. Also, quadratic relationship should be more clearly shown by plotting RO2s against reacted alpha-pinene (like Figure 1 in Zhao et al., 2018 PNAS).

Response: In the system of Zhao et al (2018), the HOM formation took place in a flow reactor (residence time around 10 s), while our measurements are over an average residence time of around 50 minutes. Therefore, we do not expect an exact quadratic (or in fact, square root) dependence as in the cited study. In addition, we only argued that the increase in radicals should be less steep than the closed shell species, due to the loss rate being proportional to $(RO2)^2$, which we believe is valid.

Although we have a limited amount of conditions, we plotted the same molecules as in Fig. 6 as a function of reacted a-pinene (see below), and it is clear that the two radicals show the least steep increase.

[Figure]

Figure R2 Normalized concentrations of radicals, monomers and dimers with the variation of reacted alpha-pinene.

6. Line 264-265. This interpretation is not accurate. The AMS measures only the elemental composition. Let's do the math here for example, if a particle-phase molecule of a Mw of 300 has one Cl atom (Mw 35.5), then the Cl/Org ~ 13%. A measured Cl/Org ~ 3% suggests about 23% of the molecules contain Cl on average. This is not a trivial fraction. Of course, this number could vary if the average Mw of particle-phase molecules change, which the authors could provide better estimates. The point is that 3% Cl/Org does not mean Cl contribute little to SOA. If considering that Cl addition could also lead to products without Cl through fragmentation, it seems the abstraction: addition occurs at 3:1 or lower fractions. Abstraction is still larger, but one cannot rule out Cl

addition. It might not produce HOM, but definitely contribute to SOA formation.

Response We agree with the reviewer that Cl does contribute to SOA, but as also the reviewer's calculation showed, the fraction of molecules containing Cl seems to be lower than those that don't, as we suggested in our text. We have now still made more clear that this value of 3% should not be overinterpreted.

7. HOM formation under high NOx. More information should be provided in this section. For example, how much O3 are formed? Given the reaction rates with ï¿A¿a-pinene, the authors could also constrain the fractions of alpha-pinene reacting with O3 vs. Cl. If alpha-pinene + O3 is the dominant process, then the HOM could be mostly from alpha-pinene + O3 and the statement in Line 287-289 might be wrong. Alternatively, if alpha-pinene + O3 is not the dominant process for alpha-pinene, then the author should consider adding alpha-pinene + O3 in equation (1) and (2) and split HOM contribution from both oxidants. In this way, if the HOM yield from alpha-pinene + O3 is known based on the authors' prior work, then the HOM yield from alpha-pinene + Cl in the presence of NOx could be estimated as well. I'm also curious whether the monomeric range mass spectrum as shown in Figure 4 changes when NOx in present (e.g., different dominating patterns).

Response: We had measurement of ozone during our experiment. As shown in figure 9, around 8, 12, and 15 ppb of ozone was formed as NO2 was introduced into the chamber. We calculated reacted alpha-pinene from ozone and Chlorine atom as following: $[ap\_ozone]/[ap\_Cl]=[AP]*[Ozone]*2.5e10*k_{ap+ozone}/[AP]*[Cl]*k_{ap+Cl}$, the results is around 0.2. with the results, we conclude that AP reaction with chlorine atom is the main reaction in the system (Line 312-315).

About the HOM yield with presence of NOx, we want to avoid speculating about exact values in

this case, in particular because the NOx is not an "on-off" effect, and the exact amount of NOx will affect the HOM yields. NOx will compete both with termination by other bimolecular pathways, like reaction with RO2, but in addition also with the autoxidation. Adding to this the uncertainty in the HOM yield from O3 + a-pinene with NOx, we cannot determine the HOM yields accurately enough.

With the presence of NOX, the monomeric range did not change compared with the spectrum without NOX, as we could see from Figure 4 and Figure 8.

8. The most interesting finding to me in this manuscript is that Cl oxidation of alpha pinene occurs mostly via abstraction, rather than Cl addition. As a general comment, I feel the authors should add some discussion of this preference. Has this been reported in prior studies? If not, then the authors should point out that this mechanistic behavior needs to be studied in detail in future work.

Response: According to our experimental results, we cannot claim that Cl oxidation of alpha-pinene occurs mostly via H-abstraction and previous study suggest that this reaction occurred via addition pathway (Cai and Griffin 2006). We have thus removed some statements in the revised version. It may be that the abstraction pathway is more efficient in producing HOM, or that a highly functionalized HOM may lose Cl through some unknown process.

Technical comments:

1. Line 39 in Abstract. Change "NO3-based" to "NO3–based" and define "CI-APi-TOF".

Response: We changed 'NO3-based' to 'Nitrate-based ' and defined 'CI-APi-TOF' as Chemical Ionization-Atmospheric Pressure Interface-Time of Flight (Line 40).

2. Line 165. NOx = NO + NO2 should be defined earlier in the manuscript.

Response: We added definition in the earlier part (Line 39).

3. Line 229-230. References are needed.

Response: We added references in the revised version (Line 245).

Reference

Cai X and Griffin R J 2006 Secondary aerosol formation from the oxidation of biogenic

hydrocarbons by chlorine atoms *Journal of Geophysical Research Atmospheres* **111** Online:

https://doi.org/10.1029/2005JD006857

Kürten A, Rondo L, Ehrhart S and Curtius J 2012a Calibration of a chemical ionization mass

spectrometer for the measurement of gaseous sulfuric acid *Journal of Physical Chemistry A*

**116** 6375–86

Kürten A, Rondo L, Ehrhart S and Curtius J 2012b Calibration of a chemical ionization mass

spectrometer for the measurement of gaseous sulfuric acid *Journal of Physical Chemistry A*

**116** 6375–86 Online: https://doi.org/10.1021/jp212123n

Zheng J, Yang D, Ma Y, Chen M, Cheng J, Li S and Wang M 2015 Development of a new corona

discharge based ion source for high resolution time-of-flight chemical ionization mass

spectrometer to measure gaseous H2SO4 and aerosol sulfate *Atmospheric Environment* **119**

167–73

Li, X., Chee, S., Hao, J., Abbatt, J. P. D., Jiang, J. and Smith, J. N.: Relative humidity effect on

the formation of highly oxidized molecules and new particles during monoterpene oxidation,

Atmospheric Chemistry and Physics, 19(3), 1555–1570, doi:10.5194/acp-19-1555-2019,

2019.

---

## Author Comment (AC2) · 13 Feb 2020

The comment was uploaded in the form of a supplement:
https://www.atmos-chem-phys-discuss.net/acp-2019-807/acp-2019-807-AC2-supplement.pdf